# Characterization of Cold-Tolerant *Saccharomyces cerevisiae* Cheongdo Using Phenotype Microarray

**DOI:** 10.3390/microorganisms9050982

**Published:** 2021-04-30

**Authors:** Kyung-Mi Jung, Jongbeom Park, Jueun Jang, Seok-Hwa Jung, Sang Han Lee, Soo Rin Kim

**Affiliations:** 1Cheongdo Peach Research Institute, Gyeongsangbuk-Do Agricultural Technology Administration, Cheongdo 38315, Korea; kmgod@korea.kr; 2School of Food Science and Biotechnology, Kyungpook National University, Daegu 41566, Korea; whdqja5743@naver.com (J.P.); hazel515@daum.net (J.J.); jshtjrghk0@gmail.com (S.-H.J.); sang@knu.ac.kr (S.H.L.)

**Keywords:** phenotype microarray, yeast, cold tolerance, galactose, GAL4

## Abstract

The cold-tolerant yeast *Saccharomyces cerevisiae* is industrially useful for lager fermentation, high-quality wine, and frozen dough production. *S. cerevisiae* Cheongdo is a recent isolate from frozen peach samples which has a good fermentation performance at low temperatures and desirable flavor profiles. Here, phenotype microarray was used to investigate industrial potentials of *S. cerevisiae* Cheongdo using 192 carbon sources. Compared to commercial wine yeast *S. cerevisiae* EC1118, Cheongdo showed significantly different growth rates on 34 substrates. The principal component analysis of the results highlighted that the better growth of Cheongdo on galactose than on EC1118 was the most significant difference between the two strains. The intact *GAL4* gene and the galactose fermentation performance at a low temperatures suggested that *S. cerevisiae* Cheongdo is a promising host for industrial fermentation rich in galactose, such as lactose and agarose.

## 1. Introduction

Phenotype microarray is an automated high-throughput technique for the phenotypic profiling of microbial isolates or communities [1,2]. Specifically, cellular growth on microplates is measured with different substrates and stress conditions. Phenotype microarray uses preconfigured sets of microplates, which makes it different from other growth-monitoring microplate readers, such as Bioscreen C [3]. Continuous growth curves from phenotype microarrays can be used to calculate the overall fitness of microbial cells, such as lag time, growth rate, and maximum density [4]. Phenotype microarray is an essential tool for microbial phenomics, in which phenotypic profiling is described on a genome-wide scale using a knockout library [5,6,7]. Yeast phenomics integrated with other omics approaches enables determining yeast phenotypic diversity and accurate phenotype prediction, including complex traits [8,9,10,11].

The industrial yeast *Saccharomyces cerevisiae* has a high level of genetic and phenotypic diversity, driven by long-term adaptation and evolution for different applications [12,13]. For example, wine industry strains are highly resistant to grape antioxidants and antimicrobial additives [14]. In contrast, the strains used in the bioethanol industry have a high ethanol productivity [14]. Hence, to isolate new yeast strains for industrial applications, it is important to establish a method for rapid screening and selecting phenotypes that are superior to existing industrial strains.

Among various industrially attractive phenotypes of *S. cerevisiae*, cold tolerance is one of the most essential traits for the beverage and food industry [15]. Wine and beer fermentation at a low temperature improves flavor by preventing spoilage microorganisms [16,17,18]. Moreover, cold-tolerant strains are required to produce unfermented frozen dough with viable cells [19]. Transcriptomics and quantitative trait loci (QTL) analysis suggested that cold tolerance is a complex trait that is difficult to engineer genetically [20,21]. As in the case of the most common lager yeast *S. pastorianus* (a hybrid of ale-type *S. cerevisiae* and cold-tolerant *S. eubayanus*), cold-tolerant *S. cerevisiae* can be achieved by prolonged adaptation to cold stress, which might accompany hybridization with other cold-tolerant wild yeast species [22]. Therefore, cold-tolerant isolates of *S. cerevisiae* could have different origins and fermentation phenotypes [23,24,25].

*S. cerevisiae* Cheongdo (KACC 93277P) is a cold-tolerant strain isolated from frozen peach samples [16]. In this study, the possibility of the industrial use of *S. cerevisiae* Cheongdo was investigated using a phenotype microarray with 192 carbon sources. The results suggested that the Cheongdo strain has a high potential for several industrial applications.

## 2. Materials and Methods

### 2.1. Strains and Fermentation

The cold-tolerant yeast *S. cerevisiae* Cheongdo [16] was compared to a commercial wine yeast *S. cerevisiae* EC1118 (LALVIN, Montreal, QC, Canada). Cells were precultured in 5 mL of YPD (1% yeast extract, 2% peptone, and 2% glucose). Flask fermentation was performed in 125 mL Erlenmeyer flasks containing 20 mL of YPD or YPG (1% yeast extract, 2% peptone, and 2% galactose) with an initial cell density of 0.01 or 0.1 OD_600_ (optical density at 600 nm). Flasks were incubated at 10 °C or 30 °C and at 130× rpm.

### 2.2. Phenotype Microarray

The Biolog PM01 and PM02 microplates were purchased from Biolog (Hayward, CA, USA). The strains were incubated overnight in 5 mL of YPD medium, then washed with distilled water. Afterwards, cells were suspended in IFY-0 base (Biolog 72231) at an initial cell density of 0.01 OD_600_. One hundred microliters of the cell suspension were inoculated into each microplate’s well; then, microplates were incubated in an OmniLog reader (Biolog). Each experiment was performed in duplicate. Each well contained a tetrazolium dye to detect color changes (colorless to red) due to cell respiration (NADH production). The color changes were automatically recorded every 15 min using a charge-coupled device camera and converted into OmniLog units [13,26]. The results were analyzed by Student’s *t*-test and principal component analysis (PCA) using R studio’s OPM (https://cran.r-project.org/src/contrib/Archive/opm/, accessed on 29 April 2021) and ggplot2 (https://cran.r-project.org/src/contrib/Archive/ggplot2/, accessed on 29 April 2021) [4,27].

### 2.3. DNA Sequencing

The genomic DNA was purified by Exgene Cell SV Kit (GeneAll, Seoul, Korea). The 2.6 kb *GAL4* gene (NCBI Gene ID: 855828) was amplified by PCR using Q5 polymerase and the Kim1012 (5′-GATGCACAGTAGAAGTGAAC-3′) and Kim1013 (5′-CATCTCCAGATTGTGTCTAC-3′) primers, and the PCR product was purified by Exgene PCR SV Kit (GeneAll, Seoul, Korea). Sanger sequencing of the gene was performed by Cosmogenetech (Seoul, Korea) using Kim1012, Kim1013, Kim1037 (5′-GTTGTAATAATTGTGCGGTC-3′), and Kim1038 (5′-CACTCACCGACGCTAATGAT-3′) primers.

### 2.4. High-Performance Liquid Chromatography (HPLC) Analysis

Glucose, galactose, and ethanol were analyzed by HPLC (Agilent Technologies, 1260 series, Karlsruhe, Germany) equipped with a Rezex-ROA Organic Acid H+ (8%) (150 mm × 4.6 mm) column (Phenomenex Inc., Torrance, CA, USA). Columns were eluted with 0.005 N H_2_SO_4_ at 50 °C, and the flow rate was set at 0.6 mL/min.

## 3. Results

### 3.1. Phenotype Microarray Analysis

The growth patterns of *S. cerevisiae* Cheongdo and EC1118 strains on the phenotype microarray with 192 carbon sources were monitored for 72 h, and some representative growth patterns (12 carbon sources) are shown in Figure 1. The growth values (y-axis) were presented as OmniLog units, representing the degree of respiration. Each growth curve showed different maximum values and rates. In particular, conventional carbon sources, such as d-mannose, d-fructose, d-glucose, d-maltose, and sucrose, supported an efficient growth of both Cheongdo and EC1118 strains. Some carbon sources, such as 2-deoxy-d-ribose, dihydroxy-acetone, l-lyxose, and pyruvic acid, yielded lower cell growth than that observed with conventional sugars for both strains. The negative control (without carbon source) was considered as the baseline respiration signal (no growth).

To better understand the phenotypic differences between Cheongdo and EC1118, the PCA of the maximum growth values (OmniLog units) on 192 carbon sources was performed. The results are presented as a biplot (Figure 2). The loadings (strains) were clearly separated, and the scores (carbon sources) were divided into three groups (A, B, and C). Group A represented carbon sources that did not support much growth for both strains. Group B represented carbon sources that showed different maximum growth values between the two strains, such as sucrose, d-glucose, and turanose. Lastly, group C represented galactose, which yielded the most significant difference between the two strains.

The detailed phenotypic differences between Cheongdo and EC1118 were compared based on the calculated growth rates (OmniLog units/h at 15 h). Table 1 shows the 34 carbon sources that yielded significantly different growth rates between the two strains (*p* < 0.05). Again, galactose resulted in the greatest difference between their growth rates; Cheongdo showed 5.6 times higher growth rates on galactose than on EC1118. EC1118 barely grew on galactose, similarly to the negative control (no carbon source). Meanwhile, the other 31 carbon sources, except for d-serine and γ-Cyclodextrin, supported higher growth rates for EC1118 than for Cheongdo strains. Although the growth on the 33 carbon sources was statistically significant, the values were close to those of the negative control, suggesting that these carbon sources cannot support the growth of both Cheongdo and EC1118 strains.

### 3.2. Validation of Phenotype Microarray Results by Flask Fermentation

To validate the growth results on the Biolog microplates of phenotype microarray, *S. cerevisiae* Cheongdo and EC1118 strains were tested by flask fermentation. As highlighted in Figure 3A, the growth on galactose was significantly different between the two strains (*p* < 0.01) on microplates, while the growth on d-glucose was not. The results obtained by flask fermentation were in accordance with the microplate results. The growth curves in glucose using flasks overlapped for both strains, suggesting that there was no difference. However, a growth difference in galactose was observed between the two strains when using flasks (Figure 3B,C); EC1118 was unable to grow on galactose, whereas Cheongdo showed significantly increased growth. For the Cheongdo strain, a difference in the growth rate between glucose and galactose was observed with flask fermentation, which was not observed with microplates. We speculated that the different culture conditions in flasks (microaerobic) and microplates (no agitation) are associated with different growth patterns.

### 3.3. Molecular Mechanism of the Difference in Galactose Fermentation

The galactose-negative phenotype of *S. cerevisiae* EC1118 and various industrial strains was previously reported, and the truncation of Gal4p, encoding a transcriptional activator of the *GAL* regulon, is responsible for the phenotype [28]. The Sanger sequencing of the *GAL4* gene confirmed an insertional mutation and early termination of the *GAL4* gene in EC1118 (Figure 4). However, Cheongdo had an intact *GAL4* gene, which is identical to that of the laboratory S288C strain, although S288C cannot metabolize galactose because of mutations in other metabolic genes [28]. The genotype of the *GAL4* gene supported the galactose-positive phenotype of *S. cerevisiae* Cheongdo, which was different from other industrial strains, including EC1118.

### 3.4. Galactose Fermentation at a Low Temperature

Our previous study reported that *S. cerevisiae* Cheongdo has an excellent glucose fermentation capability at 10°C [16]. This study also confirmed that Cheongdo was able to produce a significantly higher level of ethanol (0.60 g ethanol/L·h) than EC1118 (0.52 g ethanol/L·h) using 200 g/L of glucose at 10°C (Table 2). To test if the Cheongdo strain can ferment galactose at such a low temperature, 20 g/L and 100 g/L galactose were fermented at 10°C. As shown in Figure 5, the complete consumption of 20 g/L galactose took 72 h, and 6.8 g/L ethanol was produced (0.09 g ethanol/L·h, Table 2). At 100 g/L galactose, complete consumption was achieved in 144 h, and ethanol production and productivity were improved to 42.1 g/L and 0.29 g ethanol/L·h, respectively. Galactose fermentation at a low temperature slowed down the fermentation rate of Cheongdo by six times until the consumption of 100 g/L galactose was complete (from 24 h at 30 °C to 144 h at 10 °C). However, the ethanol titers (42.1–42.2 g/L) were not significantly affected by the fermentation temperature. From these results, it was concluded that the cold-tolerant phenotype of Cheongdo is applicable to both glucose and galactose fermentation.

## 4. Discussion

In the present study, phenotype microarray was successfully applied to screen a substrate range and growth patterns of *S. cerevisiae* Cheongdo. Comparative analysis against a well-known industrial strain EC1118 confirmed that the growth of Cheongdo on conventional carbon sources, such as glucose and sucrose, is comparable to the growth of the industrial strain (i.e., no significant difference was found). Moreover, the superior galactose-fermenting phenotype of Cheongdo was easily identified via comparative phenotype microarray, which was also found to be industrially attractive.

Several industrial and laboratory strains of *S. cerevisiae* have the galactose-negative (Gal−) phenotype [28]. The Gal− phenotype of two representative wine yeast strains, EC1118 and LalvinQA23, can be explained by mutations in the *GAL4* gene, which is a transcription factor required for *GAL* gene activation (*GAL1/2/7/10*) [29]. EC1118 and LalvinQA23 have different truncated *GAL4* alleles, and are 626 amino acids (1867 insertion A), and 465 amino acids (1354 insertion A) in length, respectively, instead of the 881 amino acids of the wild-type sequence. The expression of the wild-type *GAL4* gene in these strains recovered galactose metabolism, which confirmed that the truncated *GAL4* alleles are responsible for their Gal− phenotype. Meanwhile, a representative laboratory strain S288C has the wild-type *GAL4* gene, but it has a Gal− phenotype due to the nonsynonymous single-nucleotide polymorphisms in the *GAL* genes (*GAL1/2/10*) [30]. Although the Gal− phenotype is wildly spread among *S. cerevisiae* strains, *S. cerevisiae* Cheongdo is capable of fermenting galactose (20−100 g/L) even at a low temperature (10 °C), with ethanol productivities (0.09−0.29 g/L-h) comparable to those of glucose (0.10−0.60 g/L-h). To the best of our knowledge, this is the first report describing galactose fermentation by *S. cerevisiae* at 10 °C.

Limited information is available on the ethanol fermentation capabilities of galactose-fermenting (Gal+) *S. cerevisiae* strains. Representative laboratory strains, BY4743 and CEN.PK, showed 0.13–0.29 g/L-h ethanol productivities using 40–45 g/L galactose [31,32]. Their adaptive laboratory evolution improved ethanol productivities up to 0.68 g/L-h [33]. *S. cerevisiae* KL17 isolated from soil recorded the highest ethanol productivity (0.63–2.07 g/L-h) using 20–100 g/L galactose under highly aerobic conditions (200× rpm) with an initial cell density of 0.2–0.5 OD_600_ [34]. Given that conditions in the present study were less aerobic (130 rpm) with an lower initial cell density (0.1 OD_600_), the ethanol productivity of the Cheongdo strain (0.59–1.76 g/L-h) can be considered similar to that of the LK17 strain. Again, these results are limited to 30 °C fermentation, because there is no report of galactose fermentation at a lower temperature.

Several studies have reported metabolic engineering approaches to improve the galactose fermentation of *S. cerevisiae* [32,33,35]. The deletion of *GAL* gene repressors (*GAL6*, *GAL80*, and *MIG1*) improved the ethanol production from galactose, even though there were some growth defects [35]. An overexpression library led to the identification of truncated *TUP1* as an overexpression target to activate *GAL* genes [32]. Recently, adaptive evolution successfully improved galactose fermentation, and their mechanisms were partially explained by a mutation in the *GAL80* gene [33]. A milk-adapted natural isolate of *S. cerevisiae* showed the derepression of glucose on galactose, which was partially explained by mutations in the upstream repressing sequence (URS) sites of Gal genes, resulting in the lack of Mig1p-mediated repression [36]. Cheese strains of *S. cerevisiae* showed highly divergent alleles of high-affinity galactose transporter *GAL2*, which suggests that galactose uptake may limit galactose metabolism the most [13]. Meanwhile, the Cheongdo strain showed a similar ethanol productivity on both 20 g/L glucose and 20 g/L galactose, regardless of the fermentation temperature (Table 2). This efficient galactose fermentation capacity suggests that the *GAL* gene repressors of the Cheongdo strain may not function, which should be confirmed in a follow-up study.

The cold tolerance and galactose fermentation capacity of *S. cerevisiae* are phenotypes of particular interest in different foods and industrial fermentations. For coffee and cocoa bean fermentation, using a starter culture of *S. cerevisiae* was recently proposed for improved and consistent fermented products [37,38]. This is because the structural carbohydrates of coffee and cocoa beans are high in galactose [39,40], and their fermentation can be affected by the galactose fermentation capacity of *S. cerevisiae*. Moreover, galactose-rich food wastes, such as spent coffee grounds and cheese whey, can be transformed into value-added products by *S. cerevisiae* fermentation [41,42,43,44]. Lastly, red algae hydrolysates rich in galactose (approximately 20% of dry matter) [45] are promising alternative biomass for bioethanol and chemical production by *S. cerevisiae* [46,47]. Thus, *S. cerevisiae* Cheongdo has industrial potential that can be used for galactose-containing substrate fermentations, especially when low fermentation temperatures are preferred.

## Figures and Tables

**Figure 1 microorganisms-09-00982-f001:**
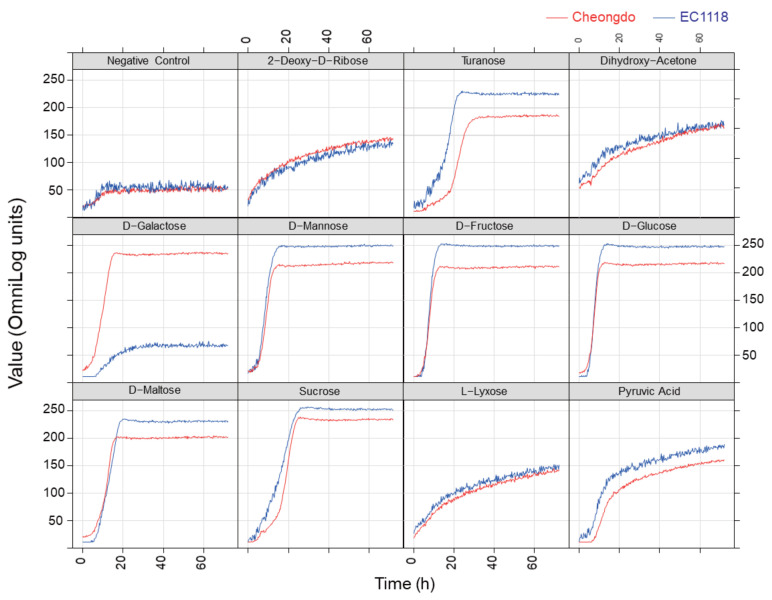
Representative growth patterns of *S. cerevisiae* Cheongdo (red) and EC1118 (blue) strains on Biolog PM01 and PM02 microplates with 192 carbon sources for 72 h at 30 °C. OmniLog units represent a measurement of dye reduction: i.e., a measurement of microbial respiration.

**Figure 2 microorganisms-09-00982-f002:**
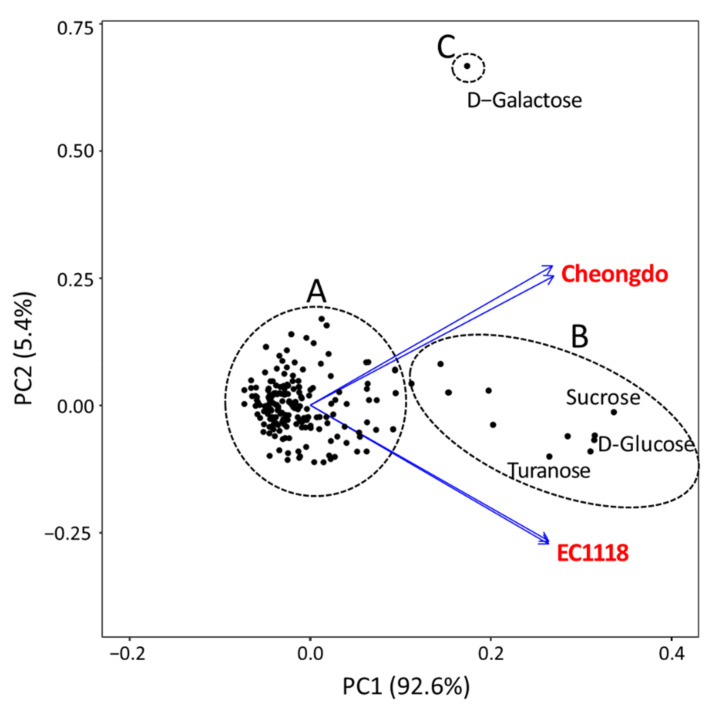
Biplot of variable loadings (represented by vectors) and scores (marked by points) from principal component analysis (PCA) of the maximum growth values (OmniLog units) of *S. cerevisiae* Chengdu and EC1118 strains on 192 carbon sources for 72 h at 30 °C.

**Figure 3 microorganisms-09-00982-f003:**
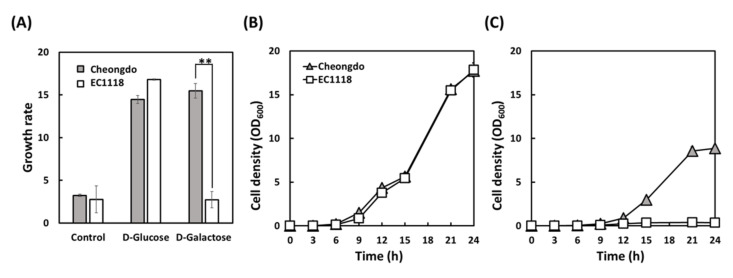
Comparison of Biolog microplates (**A**) and Erlenmeyer flasks (**B**,**C**) for the growth of *S. cerevisiae* Cheongdo and EC1118 strains using glucose (**A**,**B**) and galactose (**A**,**C**) as a sole carbon source at 30 °C. (**A**) The average growth rate (OmniLog units/h) at 15 h on microplates. (**B**,**C**) Growth profiles (OD_600_, optical density at 600 nm). Control does not contain any carbon source. Two asterisks (**) indicate a significant difference at *p* < 0.01. An initial cell density was adjusted to 0.01 OD_600_ for both microplates and flasks.

**Figure 4 microorganisms-09-00982-f004:**
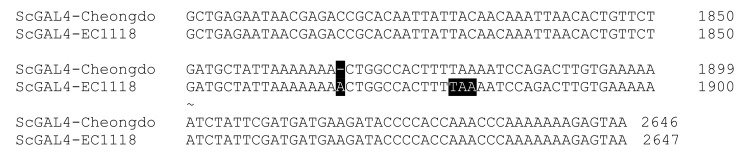
Nucleotide sequence alignment of the *GAL4* gene of *S. cerevisiae* Cheongdo (wild-type) and EC1118 (1867 insertion A) resulted in the frameshift and an early termination at TAA (highlighted in black; truncated Gal4p from 881 to 626 amino acids).

**Figure 5 microorganisms-09-00982-f005:**
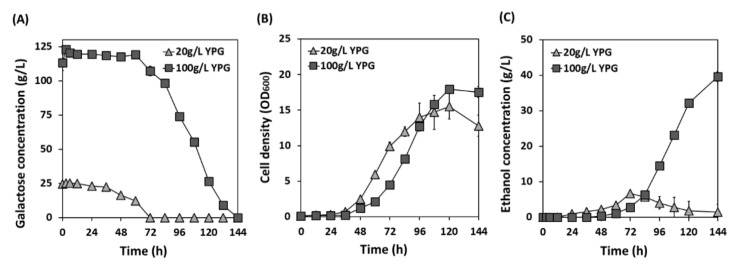
Galactose fermentation capability of cold-tolerant *S. cerevisiae* Cheongdo at 10 °C. Galactose concentration (**A**), cell density (**B**), and ethanol concentration (**C**) were compared using galactose (20 g/L, 100 g/L) as a sole carbon source. Fermentation was performed at 10 °C and 130× rpm with an initial cell density of 0.1 OD_600_.

**Table 1 microorganisms-09-00982-t001:** Significantly different growth rates ^1^ of *S. cerevisiae* Cheongdo and EC1118 strains on different carbon sources.

Carbon Source ^2^	Cheongdo	EC1118	Difference	*p*-Value ^3^
Negative Control	3.23 ± 0.14	2.77 ± 1.56	0.47	0.74
d-Galactose	15.47 ± 0.85	2.73 ± 0.94	12.73	0.01
l-Pyroglutamic Acid	1.43 ± 0.33	4.83 ± 0.24	3.40	0.01
3-Hydroxy-2-Butanone	2.93 ± 0.19	5.53 ± 0.28	2.60	0.01
N-Acetyl-l-Glutamic Acid	1.33 ± 0.19	3.73 ± 0.28	2.40	0.02
2,3-Butanedione	1.87 ± 0.19	4.27 ± 0.38	2.40	0.03
Butyric Acid	2.30 ± 0.14	4.57 ± 0.33	2.27	0.04
Palatinose	1.97 ± 0.24	4.00 ± 0.19	2.03	0.01
d,l-Carnitine	1.33 ± 0.09	3.33 ± 0.19	2.00	0.02
Putrescine	1.03 ± 0.33	2.83 ± 0.33	1.80	0.03
3-0-β-d-Galactopyranosyl-d-Arabinose	1.30 ± 0.14	3.07 ± 0.19	1.77	0.01
l-Phenylalanine	1.20 ± 0.19	2.97 ± 0.24	1.77	0.02
d-Serine	2.67 ± 0.00	1.00 ± 0.09	1.67	0.03
β-Hydroxy Butyric Acid	1.20 ± 0.09	2.73 ± 0.19	1.53	0.02
l-Lysine	1.43 ± 0.05	2.90 ± 0.05	1.47	0.00
β-Methyl-d-Xyloside	2.07 ± 0.09	3.53 ± 0.00	1.47	0.03
Glycine	0.80 ± 0.00	2.23 ± 0.05	1.43	0.01
l-Alaninamide	1.27 ± 0.19	2.67 ± 0.28	1.40	0.04
l-Glutamic Acid	3.43 ± 0.24	4.80 ± 0.19	1.37	0.03
l-Arginine	1.37 ± 0.24	2.73 ± 0.19	1.37	0.03
d-Ribono-1,4-Lactone	0.77 ± 0.05	2.10 ± 0.14	1.33	0.03
d-Tartaric Acid	1.63 ± 0.05	2.90 ± 0.05	1.27	0.00
Hydroxy-l-Proline	1.53 ± 0.09	2.77 ± 0.14	1.23	0.01
Sorbic Acid	1.27 ± 0.09	2.47 ± 0.19	1.20	0.03
γ-Cyclodextrin	3.03 ± 0.24	1.87 ± 0.19	1.17	0.04
N-Acetyl-d-Glucosaminitol	0.83 ± 0.14	1.97 ± 0.24	1.13	0.04
3-Methyl Glucose	2.27 ± 0.09	3.30 ± 0.05	1.03	0.02
l-Homoserine	0.80 ± 0.09	1.83 ± 0.05	1.03	0.02
δ-Amino Valeric Acid	1.23 ± 0.05	2.23 ± 0.05	1.00	0.00
d-Fucose	2.23 ± 0.05	3.20 ± 0.00	0.97	0.02
Sebacic Acid	1.27 ± 0.09	2.20 ± 0.09	0.93	0.01
l-Methionine	1.43 ± 0.05	2.30 ± 0.05	0.87	0.00
Citraconic Acid	1.93 ± 0.19	2.77 ± 0.14	0.83	0.04
γ-Amino Butyric Acid	1.33 ± 0.00	1.97 ± 0.05	0.63	0.03
l-Sorbose	1.60 ± 0.09	2.20 ± 0.09	0.60	0.02

^1^ The average growth rate was calculated as OmniLog units/h at 15 h. ^2^ In descending order of growth differences. ^3^ *p* < 0.05.

**Table 2 microorganisms-09-00982-t002:** Comparison of ethanol productivity (g/L-h) of cold-tolerant *S. cerevisiae* Cheongdo under various conditions

Carbon Source ^1^	10 °C ^2^		30 °C ^3^	
	Cheongdo	EC1118	Cheongdo	EC1118
d-Glucose 20 g/L	0.10 ± 0.00	0.11 ± 0.00	0.66 ± 0.06	0.55 ± 0.02
d-Glucose 200 g/L	0.60 ± 0.00 *	0.52 ± 0.00	3.33 ± 0.01	3.38 ± 0.03
d-Galactose 20 g/L	0.09 ± 0.01	No growth	0.59 ± 0.01	No growth
d-Galactose 100 g/L	0.29 ± 0.00	No growth	1.76 ± 0.17	No growth

* Significant difference between Cheongdo and EC1118 under the same conditions (*p* < 0.05). ^1^ Fermentation was performed at 130× rpm with an initial cell density of 0.1 OD_600_. ^2^ The average ethanol productivity at 10 °C was calculated at 72 h and 144 h for 20 g/L and 200 (100) g/L sugars, respectively. ^3^ The average ethanol productivity at 30 °C was calculated at 12 h and 24 h for 20 g/L and 200 (100) g/L sugars, respectively.

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
