# Peer review of "Characterization of Cold-Tolerant Saccharomyces cerevisiae Cheongdo Using Phenotype Microarray"

_microorganisms, 2021, doi:10.3390/microorganisms9050982_

Round 1

Reviewer 1 Report

The manuscript by Jung, Park and colleagues describes the growth characterization of the S. cerevisae Cheongdo strain, which offers promise as a interesting alternative/supplement to the existing and currently used industrial strains. The work is well-written, methodologically sound and rigorous and the results described in detail. The authors confirmed the most significant growth difference between Cheongdo and EC1118 (D-galactose) on microplates using growth in flasks, and further characterized it more precisely in terms of ethanol production, which is very commendable. I do not have any major issues with this work, it looks well, it reads well, and it is very solid. Perhaps one thing to consider would be to include a direct growth comparison between Cheongdo and another Gal+ S. cerevisiae strain, but I would not hold the paper back just for that.

Author Response

The manuscript by Jung, Park and colleagues describes the growth characterization of the S. cerevisae Cheongdo strain, which offers promise as a interesting alternative/supplement to the existing and currently used industrial strains. The work is well-written, methodologically sound and rigorous and the results described in detail. The authors confirmed the most significant growth difference between Cheongdo and EC1118 (D-galactose) on microplates using growth in flasks, and further characterized it more precisely in terms of ethanol production, which is very commendable. I do not have any major issues with this work, it looks well, it reads well, and it is very solid.

Response: We sincerely appreciate the reviewer for thoroughly evaluating our manuscript.

Perhaps one thing to consider would be to include a direct growth comparison between Cheongdo and another Gal+ S. cerevisiae strain, but I would not hold the paper back just for that.

Response: We thank the reviewer for this valuable suggestion. As suggested, the following paragraph was added to describe other Gal+ strains and their ethanol productivity on galactose:

Page 8, Line 227–237: Limited information is available on the ethanol fermentation capabilities of galac-tose-fermenting (Gal+) S. cerevisiae strains. Representative laboratory strains, BY4743 and CEN.PK, showed 0.13–0.29 g/L-h ethanol productivities using 40–45 g/L galactose [31,32]. Their adaptive laboratory evolution improved ethanol productivities upto 0.68 g/L-h [33]. S. cerevisiae KL17 isolated from soil recorded the highest ethanol productivity (0.63–2.07 g/L-h) using 20–100 g/L galactose under highly aerobic conditions (200 rpm) with an ini-tial cell density of 0.2–0.5 OD600 [34]. Given that conditions in the present study were less aerobic (130 rpm) with lower initial cell density (0.1 OD600), the ethanol productivity of the Cheongdo strain (0.59–1.76 g/L-h) can be considered similar to that of the LK17 strain. Again, these results are limited to 30°C fermentation because there is no report of galactose fermentation at a lower temperature.

Reviewer 2 Report

The manuscript by Jung et al entitled "Characterization of Cold-Tolerant Saccharomyces cerevisiae Cheongdo using Phenotype Microarray" describes a comparison between S.cerevisiae isolate from frozen peach and a well known wine strain, EC1118.

In my opinion, a comparison between two S.cerevisiae strains in what concerns industrially relevant phenotypes does not form a sufficiently complete body of work to justify publication, unless a new striking phenotype is uncovered and its genetic underpinnings described.

In the present manuscript, the authors enphasize the excellent performance of Cheongdo strain on galactose, which is not a new phenotype in S. cerevisiae (see 10.1016/j.cub.2019.02.038 and 10.1093/molbev/msy066). The relevant comparison would be with strains from a lineage previously described in these papers that shows improved galactose metabolism, but imagining they were unable to obtain the relevant strains, at the very least, the authors should discuss the findings in  these papers (they refer one of them but not in relation to galactose metabolism) and compare their findings with those already published.

For these reasons I do not find this manuscript worthy of publication without major remodelling of the study design and contextualization of the work.

Author Response

The manuscript by Jung et al entitled "Characterization of Cold-Tolerant Saccharomyces cerevisiae Cheongdo using Phenotype Microarray" describes a comparison between S. cerevisiae isolate from frozen peach and a well known wine strain, EC1118.

Response: We sincerely appreciate the reviewer for their review of our manuscript.

In my opinion, a comparison between two S. cerevisiae strains in what concerns industrially relevant phenotypes does not form a sufficiently complete body of work to justify publication, unless a new striking phenotype is uncovered and its genetic underpinnings described.

Response: We appreciate the reviewer’s critical concern. We would like to point out that in addition to the new phenotypes of the Cheongdo strain, another significant aspect of the study benefits readers in this field of study. In particular, this manuscript describes detailed procedures and statistical analysis for phenotype microarray, which will be useful for other researchers attempting to comprehensively analyze new yeast isolates.

In the present manuscript, the authors enphasize the excellent performance of Cheongdo strain on galactose, which is not a new phenotype in S. cerevisiae (see 10.1016/j.cub.2019.02.038 and 10.1093/molbev/msy066).

Response: We thank the reviewer for their valuable comment. The above references were discussed in the revised manuscript as follows:

Page 8, Line 244–249: A milk-adapted natural isolate of S. cerevisiae showed the derepression of glucose on galactose, which was partially explained by mutations in the upstream repressing sequence (URS) sites of Gal genes, resulting in the lack of Mig1p-mediated repression (Duan et al., 2019). Cheese strains of S. cerevisiae showed highly divergent alleles of high-affinity galactose transporter GAL2, which suggests that galactose uptake may most limit galactose metabolism (Legras et al., 2018).

The relevant comparison would be with strains from a lineage previously described in these papers that shows improved galactose metabolism, but imagining they were unable to obtain the relevant strains, at the very least, the authors should discuss the findings in these papers (they refer one of them but not in relation to galactose metabolism) and compare their findings with those already published. For these reasons I do not find this manuscript worthy of publication without major remodelling of the study design and contextualization of the work.

Response: We thank the reviewer for their valuable suggestions. As suggested, the following paragraph was added to discuss galactose fermentation results in prior studies:

Page 8, Line 227–237: Limited information is available on the ethanol fermentation capabilities of galac-tose-fermenting (Gal+) S. cerevisiae strains. Representative laboratory strains, BY4743 and CEN.PK, showed 0.13–0.29 g/L-h ethanol productivities using 40–45 g/L galactose [31,32]. Their adaptive laboratory evolution improved ethanol productivities upto 0.68 g/L-h [33]. S. cerevisiae KL17 isolated from soil recorded the highest ethanol productivity (0.63–2.07 g/L-h) using 20–100 g/L galactose under highly aerobic conditions (200 rpm) with an ini-tial cell density of 0.2–0.5 OD600 [34]. Given that conditions in the present study were less aerobic (130 rpm) with lower initial cell density (0.1 OD600), the ethanol productivity of the Cheongdo strain (0.59–1.76 g/L-h) can be considered similar to that of the LK17 strain. Again, these results are limited to 30°C fermentation because there is no report of galactose fermentation at a lower temperature.

 New references

  1. GOSHIMA, T.; TSUJI, M.; INOUE, H.; YANO, S.; HOSHINO, T.; MATSUSHIKA, A. Bioethanol Production from Lignocellulosic Biomass by a Novel Kluyveromyces marxianus Strain. Bioscience, Biotechnology, and Biochemistry 2013, 77, 1505-1510, doi:10.1271/bbb.130173.
  2. Kim, J.H.; Ryu, J.; Huh, I.Y.; Hong, S.-K.; Kang, H.A.; Chang, Y.K. Ethanol production from galactose by a newly isolated Saccharomyces cerevisiae KL17. Bioprocess and Biosystems Engineering 2014, 37, 1871-1878, doi:10.1007/s00449-014-1161-1.
  3. Duan, S.-F.; Shi, J.-Y.; Yin, Q.; Zhang, R.-P.; Han, P.-J.; Wang, Q.-M.; Bai, F.-Y. Reverse Evolution of a Classic Gene Network in Yeast Offers a Competitive Advantage. Current Biology 2019, 29, 1126-1136.e1125, doi:https://doi.org/10.1016/j.cub.2019.02.038.
  4. Legras, J.-L.; Galeote, V.; Bigey, F.; Camarasa, C.; Marsit, S.; Nidelet, T.; Sanchez, I.; Couloux, A.; Guy, J.; Franco-Duarte, R., et al. Adaptation of S. cerevisiae to Fermented Food Environments Reveals Remarkable Genome Plasticity and the Footprints of Domestication. Mol Biol Evol 2018, 35, 1712-1727, doi:10.1093/molbev/msy066.